# Assessing the Load, Virulence and Antibiotic-Resistant Traits of ESBL/Ampc *E. coli* from Broilers Raised on Conventional, Antibiotic-Free, and Organic Farms

**DOI:** 10.3390/antibiotics11111484

**Published:** 2022-10-26

**Authors:** Silvia Tofani, Elisa Albini, Francesca Blasi, Lucilla Cucco, Carmela Lovito, Carmen Maresca, Michele Pesciaroli, Serenella Orsini, Eleonora Scoccia, Giovanni Pezzotti, Chiara Francesca Magistrali, Francesca Romana Massacci

**Affiliations:** 1Istituto Zooprofilattico Sperimentale dell’Umbria e delle Marche “Togo Rosati”, 06126 Perugia, Italy; 2Istituto Zooprofilattico Sperimentale del Lazio e della Toscana “M. Aleandri”, 00178 Roma, Italy; 3Istituto Zooprofilattico Sperimentale della Lombardia e Dell’Emilia-Romagna “Bruno Ubertini”, 25124 Brescia, Italy

**Keywords:** extended-spectrum cephalosporins, livestock, whole genome sequencing

## Abstract

Poultry is the most likely source of livestock-associated Extended Spectrum Beta-Lactamase (ESBL) and plasmid-mediated AmpC (pAmpC)-producing *E. coli* (EC) for humans. We tested the hypothesis that farming methods have an impact on the load of ESBL/pAmpC-EC in the gut of broilers at slaughter. Isolates (*n* = 156) of antibiotic-free (AF), organic (O), and conventional (C) animals were characterized for antibiotic susceptibility and antibiotic resistance genes. Thirteen isolates were whole-genome sequenced. The average loads of ESBL/pAmpC-EC in cecal contents were 4.17 Log CFU/g for AF; 2.85 Log CFU/g for O; and 3.88 Log CFU/g for C type (*p* < 0.001). ESBL/pAmpC-EC isolates showed resistance to antibiotic classes historically used in poultry, including penicillins, tetracyclines, quinolones, and sulfonamides. Isolates from O and AF farms harbored a lower proportion of resistance to antibiotics than isolates from C farms. Among the determinants for ESBL/pAmpC, CTX-M-1 prevailed (42.7%), followed by TEM-type (29%) and SHV (19.8%). Avian pathogenic *E. coli* (APEC), belonging to ST117 and ST349, were identified in the collection. These data confirm the possible role of a broiler as an ESBL/AmpC EC and APEC reservoir for humans. Overall, our study suggests that antibiotic-free and organic production may contribute to a reduced exposure to ESBL/AmpC EC for the consumer.

## 1. Introduction

Extended-spectrum β-lactamase (ESBL) and AmpC β-lactamase (AmpC)-producing *Escherichia coli* (ESBL/AmpC-EC) isolates are resistant to extended-spectrum cephalosporins (ESC), a class of antibiotics classified as highest-priority critically important antimicrobials (HPCIA) by the WHO [1,2]. Resistance to ESCs is linked to the presence of genetic determinants belonging to the TEM, SHV, and CTX-M families. OXA-CMY and other families are responsible for resistance to ESBL and AmpC enzymes [3,4]. Such determinants are mostly localised on mobile genetic elements that can be transferred among different lineages of the same bacterial species or between different species [4]. Therefore, the presence of ESBL/AmpC-EC in livestock is a major public health concern, not only with regards to the direct transmission of bacteria from animals to humans, but also with regards to the possible transfer of resistance determinants from commensal to zoonotic bacteria in the gut flora [3]. Among livestock production chains, poultry is the most likely livestock-associated ESBL/AmpC-EC reservoir relevant to human health [5].

Apart from being resistant to third-generation cephalosporin, ESBL/AmpC-EC isolates from poultry are often multi-resistant and potentially pathogenic to animals and humans, as they often bear virulence factors that are typical of extra-intestinal pathogenic *E. coli* (ExPEC), avian pathogenic *E. coli* (APEC), atypical enteroaggregative *E. coli* (aEAEC), and uropathogenic *E. coli* (UPEC) [6,7]. Studies in animal models have revealed APEC as a cause of urinary tract infections and meningitis, similar to those caused by human ExPEC [8]. Other studies have reported overlapping characteristics between ExPEC and APEC during cryptic outbreaks [9]. The coexistence of virulence factors and antibiotic resistance determinants in the same ESBL/AmpC-EC clone poses an additional threat to public health. However, few studies have focused on the occurrence of ESBL/AmpC-EC harbouring ExPEC/APEC virulence factors in the broiler production chain [4,10]. Virulence factors associated with APEC strains include adhesins encoded by a temperature-sensitive hemagglutinin gene (*tsh*), protectins encoded by an increased serum survival gene (*iss*), toxins encoded by an enteroaggregative toxin gene (*astA*) [11,12]. In Europe, the prevalence of ESBL/AmpC-EC-positive broilers at slaughterhouses varies broadly across countries, and is the highest in Italy (above 60%) [13]. In the last three years, however, the proportion of ESBL/AmpC-EC has slightly declined as a possible consequence of the reduction in antibiotic use [13]. In Italy, the overall sales of antibiotics in the livestock sector was almost halved from 2010 to 2017 [14]. The poultry sector widely contributed to this achievement: a strategic plan implemented by poultry producers resulted in an estimated reduction of 87% in antibiotic consumption from 2001 to 2020 on broiler farms [15]. In the same period, production characterised by reduced or no antibiotic use, including organic and antibiotic-free production, surged, particularly in the broiler sector [16]. These data suggest that organic and antibiotic-free production is effective at reducing the prevalence of antibiotic-resistant bacteria in animals [16,17]. Different studies have reported that *E. coli* strains resistant to cefotaxime are significantly more abundant in conventional chicken samples than in organic and antibiotic-free samples, supporting the hypothesis that the use of antibiotics can exert selective pressure on the microbial community [18]. Moreover, conventional production is associated with the presence of *E. coli* resistant to several antimicrobials, showing a multi-resistant profile [18]. A recent clonal spread was associated with an enhanced virulence potential in *E. coli* ST131 and ST648 strains due not only to the carriage of AMR genes but also to other fitness factors [19,20]. 

The hypothesis of this study was that the type of production has an impact on the prevalence and load of ESBL/AmpC-EC in the gut of broilers at slaughter. The isolates were described in terms of antibiotic resistance, and we further investigated a subset of isolates in terms of ST and virulence types using whole genome sequencing (WGS) to elucidate their potential for zoonotic infection. 

## 2. Materials and Methods

### 2.1. Sampling

The sampling was performed as previously described from the same research group [16]. Briefly, from February 2017 to January 2018, 809 cecal contents belonging from different production systems farms were processed: antibiotic-free (AF, *n* = 292), organic (O, *n* = 246) and conventional (C, *n* = 271). The samples were from 11 (AF), 10 (C) and 9 (O) different productive farm systems. 

### 2.2. Bacteriological Culture

Quantitative culture was used to determine the loads of *E. coli* resistant to cefotaxime (*E. coli*^cef^). Samples were processed following the procedure described by Duse et al. [21]. Briefly, 5 g of each cecal content were homogenized into a Stomacher bag (Seward Ltd., Easting Close, Worthing, West Sussex, United Kingdom) in 45 mL of 0.9% saline. The suspension (10^−1^ dilution *w*/*v*) was further 10-fold diluted in 0.9% saline, from 10^−1^ to 10^−8^. The colony-forming units (CFU) of *E. coli*^cef^ was estimated by plating 100 μL of each dilution on MacConkey agar supplemented with cefotaxime (Sigma Aldrich- Merck KGaA, Darmstadt, Germany) (1µg/mL -MacConkey^cef^). For the MacConkey^cef^ medium, the quality control was carried out according to the guidelines of the European Reference Centre for antimicrobial resistance [22], using *Salmonella* O:6,7 WHO S-17.8 as the positive control strain and *E. coli* ATCC 25922 as the negative control strain. Plates were incubated at 37 °C overnight. The number of *E. coli* CFU in the sample was estimated by counting pink or red colonies with a morphology resembling *E. coli* on MacConkey^cef^. To calculate CFU/g, the average number of colonies was multiplied by the dilution factor. For each sample, a colony was isolated from MacConkey^cef^ agar plates, incubated at 37 °C for 24 h and confirmed as belonging to *E. coli* species using oxidase, triple sugar iron and indole test. Counts of *E. coli*^cef^ were estimated following the same procedure for MacConkey^cef^ plates, respectively. For each flock, five colonies from MacConkey^cef^ were tested using agar diffusion test to confirm resistance to cefotaxime. The proportion of *E. coli*^cef^ of the total *E. coli* population in each cecal content was calculated dividing the counts of *E. coli*^cef^ for the counts of *E. coli* of the same sample. 

### 2.3. Antimicrobial Susceptibility and ESBL/Ampc Phenotype 

The 156 *E. coli*^cef^ isolates were tested for susceptibility to antimicrobial drugs by using the agar diffusion method on Mueller Hinton Agar (Oxoid Ltd., Cambridge, UK), according to the European Committee on Antimicrobial Susceptibility Testing (EUCAST) guidelines [23]. *E. coli* ATCC 25922 was used as a quality control strain. The antimicrobial discs (Oxoid Ltd) were ampicillin (10 µg), amoxicillin/clavulanic acid (30 μg), cefotaxime (30 µg), cefazolin (30 µg), chloramphenicol (30 µg), ciprofloxacin (5 µg), gentamicin (10 µg), kanamycin (30 µg), imipenem (10 µg), nalidixic acid (30 µg), streptomycin (10 µg), sulfonamides (300 µg), tetracycline (30 µg) and sulfamethoxazole/trimethoprim (25 µg). The size of inhibition diameters were interpreted following the EUCAST breakpoint tables [24]. Since the EUCAST does not provide breakpoints for cefazolin, kanamycin, nalidixic acid, sulfonamides, tetracycline and sulfamethoxazole/trimethoprim, values of the Clinical & Laboratory Standards Institute were used for these molecules [25]. The isolates were classified as Multidrug-Resistant (MDR) according to previously described criteria [26]. 

*E. coli* ATCC 25922 was used as the quality control strain. Briefly, an isolate was classified as MDR when it exhibited resistance to at least five antibiotics representing aminopenicillins (ampicillin), first-generation cephalosporins (cefazolin), third generation cephalosporins (cefotaxime), amphenicols (chloramphenicol), quinolones (nalidixic acid), fluoroquinolones (ciprofloxacin), sulfonamides, aminoglycosides (gentamycin), and tetracycline.

According to EUCAST guidelines, the isolates were considered as ESBL producing *Enterobacteriaceae* and/or AmpC β-lactamase-producing *Enterobacteriaceae* using the double disk synergy test (DDST) with a disc of cefotaxime (30 μg) and a disc of amoxicillin-clavulanate (containing 10 μg of clavulanate), positioned at a distance of 30 mm (center to center), according to EUCAST guidelines [27].

Finally, susceptibility to colistin was assessed using the broth microdilution method in order to determine the minimal inhibitory concentration (MIC), using polystyrene microtiter plates (LP Italiana SpA, Milan, Italy) and sulfate salt of colistin (Sigma Aldrech SRL, Milan, Italy), according to the EUCAST recommendations [28]. *E. coli* strains ATCC 25922 and ZTA14/0097EC [a kind gift from Professor Lucas Dominguez Rodriguez, Centro de Vigilancia Sanitaria Veterinaria (VISAVET), Universidad Complutense, Madrid, Spain] served as the quality-control strains. The classification of isolates as resistant was based on MIC values using the EUCAST criteria (resistant: MIC > 2 mg/L). 

### 2.4. DNA Extraction

For the DNA extraction of bacterial isolates, the QIAamp DNA Mini Kit (Qiagen, Hombrechtikon, Switzerland) was used following the manufacturer’s instructions.

### 2.5. Determination of Phylogenetic Groups

*E. coli* isolates were tested by PCR for characterization of the phylogenetic groups A, B1, B2, C, D, E, and F, according to Clermont et al. [29]. 

### 2.6. Determination of ESBL- and AmpC-Associated Genes 

The presence of AR genes was determined on isolates classified as ESBL and/or AmpC. For ESBL-enzymes detection, two multiplex PCRs and one simplex PCR were performed in this study: a *bla*_TEM_/*bla*_SHV_/*bla*_OXA-1-like_ multiplex PCR; a *bla*_CTX-M_ multiplex PCR including phylogenetic groups 1, 2 and 9; a *bla*_CTX-M-8/-25_ simplex PCR [30]. For AmpC-enzymes ACC, FOX, MOX, DHA, CIT and EBC one multiplex PCR was performed as previously described [31]. The presence of the allelic variant *bla*_CTXM-15_ was then investigated in all isolates positive for *bla*_CTX-M_ group 1 [32] and all the isolates positive for *bla*_CIT_-group were further investigated for *bla*_CMY-2_ group using a previously described method [33]. 

### 2.7. Whole Genome Sequencing 

In order to investigate sequence type, serotype, virulence profile, and antimicrobial resistant genes, 13 *E. coli* isolates were WG sequenced. The subset isolates were selected among the F and D phylogenetic group isolates, which are known to be pathogen for humans. 

Genomic DNAs of the pure *E. coli* cultures were extracted from 1 mL of logarithmic phase broth cultures using QIAamp DNA Mini Kit (Qiagen Inc., Hilden, Germany) following the manufacturer’s protocol for Gram-negative bacteria organisms. Each sample was then quantified with the Qubit fluorometer (QubitTM DNA HS Assay, Life Technologies, Thermo Fisher Scientific Inc., Milan, Italy). Library preparation was obtained using the Nextera XT Library Prep kit (Illumina Inc., San Diego, CA, USA) according to the manufacturer’s manual. The prepared libraries were loaded onto NextSeq 500/550 Mid Output Reagent Cartridge v2, 300 cycles kit (Illumina Inc., San Diego, CA, USA) and then sequenced on an Illumina NextSeq 500 platform to generate 150 bp paired-end reads. 

### 2.8. Sequence Analysis

Raw data were checked for quality, trimmed using Trimmomatic v0.36 (Bolger et al., 2014) and assembled using SPAdes genome assembler v3.11.1 [34]. Genomes were annotated using Prokka [35] and a maximum likelihood phylogenetic tree, based on the final alignment of core genome from Roary analysis, was constructed using FastTree 2.1.11 [36]. Manual annotation of the tree was performed in iTOL (v.5.7, https://itol.embl.de/ accessed on 10 February 2022) [37].

The bioinformatics analysis was carried out using the services of the Centre for Genomic Epidemiology (CGE), Technical University of Denmark (DTU, https://cge.cbs.dtu.dk/services/ accessed on 9 November 2021) [38,39,40,41,42]. Briefly, the fasta files were analyzed using the following CGE databases: MLST, SeroTypeFinder, ResFinder for the acquired antibiotic resistance genes, and VirulenceFinder for identifying the putative virulence factors of isolates. Based on the virulence genes described, the 13 isolates were screened for the presence of APEC virulence genes. Strains were classified as APEC, a subtype of ExPEC pathotype, when at least four among *iroN*, *iutA*, *iss*, *ompT*, and *hlyF* genes were present [43]. Moreover, we tested on our isolates the refined definition of APEC according to Johnson T. et al. [44]. Furthermore, we tested our fasta files using the ClermonTyper [45], a user-friendly tool for *Escherichia* species/phylogenetic group identification (http://clermontyping.iame-research.center/ accessed on 9 November 2021). 

### 2.9. Statistical Analyses

The bacterial counts were converted to Log CFU/g of cecal content for statistical analysis. The difference in bacterial loads from animals from the three production types were evaluated by using the Kruskal–Wallis test, after assessing the normality of data by using the Shapiro–Wilk method. The analyses were performed by using Stata 11.2 (StataCorp, College Station, TX, USA).

The difference in the proportion of isolates between animals from the three production lines was evaluated by using Pearson’s χ2 or Fisher’s test, with a significance threshold of *p* ≤ 0.05. The strength of the association was evaluated by using the odds ratio (OR), with the conventional production type as a reference (OR = 1). The analysis was performed using R (version 4.0.2, access date: 14 July 2020) package epiR [46]. To show the distribution of the putative virulence genes across the pathotypes, we performed Kruskal–Wallis rank sum testing. To investigate the distribution of genes encoding putative virulence factors, we constructed a heat map based on the distance metric “euclidean” and complete linkage method. 

## 3. Results

### 3.1. E. coli loads in the Three Management Systems

A total of 809 fecal samples from broiler cecal content at a slaughterhouse in Umbria, Italy, were collected. Samples were from three production types of breeding: conventional (C; N = 10), organic (O; N = 9) and antibiotic-free (AF; N = 11). 

A total of 156 isolates were cefotaxime-resistant after culturing on MacConkey^cef^ and they were further classified as *E. coli*. As showed in Appendix A, fifty-five isolates belonged to AF production type (35%), 47 to organic production type (30%) and 54 to conventional type (35%). 

The average *E. coli* cell count was 1.81 × 10^8^, with a ± 6.1 × 10^8^ CFU/g standard deviation (SD) cecal content, for AF animals; 1.42 × 10^8^ (SD ± 3.54 × 10^8^) CFU/g cecal content for O farm animals; and 5.31 × 10^8^ (SD ± 1.61 × 10^9^) CFU/g cecal content, for conventionally raised animals. The average *E. coli*^cef^ cell count was 5.4 × 10^6^, with a ± 2.5 × 10^7^ CFU/g standard deviation (SD) cecal content, for AF animals; 1.9 × 10^6^ (SD ± 1.2 × 10^7^) CFU/g cecal content for O farm animals; and 2 × 10^6^ (SD ± 1 × 10^7^) CFU/g cecal content, for conventionally raised animals. After Log transformation, the average *E. coli* loads was 7.47, with a SD of ± 0.85 Log CFU/g for AF animals; 7.45 (SD ± 0.91) Log CFU/g cecal content for O farm animals; and 7.69 (SD ± 1.31) Log CFU/g cecal content, for conventionally raised animals resulting in significant difference (*p* = 0.0001; Figure 1). The average *E. coli*^cef^ loads (and standard deviation) were determined to be 4.17 (SD ± 1.98) Log CFU/g cecal content, for AF animals; 2.85 (SD ± 2.16) Log CFU/g cecal content for O farm animals; and 3.88 (SD ± 2.24) Log CFU/g cecal content, for conventionally raised animals (*p* = 0.001; Figure 1). 

### 3.2. Antimicrobial Susceptibility Testing Results

Fifteen antimicrobial molecules were tested. All isolates were susceptible to colistin and imipenem. The proportion of isolates resistant to the other tested antibiotics is shown in Table 1. Antibiotic resistance equal to or higher than 50% was found for ampicillin, first generation cephalosporins (cefazolin), sulfonamides and tetracycline, independently from the production type. Resistance equal to or higher than 50% for amoxicillin/clavulanic acid and cefotaxime was found in organic and conventional production lines, for nalidixic acid in antibiotic-free and organic lines and for sulphametoxazole + trimethoprim only for conventional the production line. A resistance to colistin was not detected. The percentages of isolates resistant to chloramphenicol, ciprofloxacin, streptomycin, sulphonamides and trimethoprim + sulfamethoxazole were different among the three groups (Pearson χ2: *p* < 0.05). The organic production type was a protective factor compared to the conventional production type for chloramphenicol and sulphonamides (OR 0.20; 95%CI 0.07–0.54), while belonging to the antibiotic-free type represented a protective factor for ciprofloxacin, sulphonamides and sulfamethoxazole +trimethoprim resistances (Table 2). For the other antibiotic molecules, there were no differences in the percentages of resistance among the three production types (Pearson χ2: *p* > 0.05). 

All isolates were classified as MDR, because of resistance to three or more antimicrobial classes.

### 3.3. Molecular Analyses

The presence of genetic determinants was investigated for all 156 isolates. Phylogenetic group A was the most prevalent (90 isolates, 57.7%), followed by groups B1 (31 isolates, 19.9%), F (11 isolates, 7.0%), D (10 isolates, 6.4%), E (9 isolates, 5.8%) and C (3 isolates, 1.9%). According to Clermont O. et al. [29], 2 (1.3%) isolates were classified as “unknown”. The distribution of the phylogenetic group among the three groups did not differ (Pearson χ2: *p* > 0.05).

ESBLs and/or AmpC β-lactamases were detected by PCR in 156 isolates. The results are summarized in Table 3. According to the distribution of resistance genes, 131 (84%) isolates were phenotypically ESBL-producing *E. coli*, 18 (11%) were AmpC producers, 3 (2%) were both ESBL and AmpC producers, while 4 (3%) were classified as negative, according to EUCAST, 2018 (EUCAST, 2018). 

ESBL-producing isolates belonged to CTX-M group 1 (42.7%), even in combination with other enzymes, in particular with TEM-type (29%), followed by SHV alone (19.8%). All CTX-M group 1 positive isolates, except three, belonged to the CTX-M-15 allelic variant. The CTX-M-1^+^ isolates were not randomly distributed among the groups (Pearson χ2: *p* = 0.0013): belonging to an antibiotic-free system was a protective factor (OR 0.23, 95%CI 0.10–0.54), while isolates from organic farms were not different from the conventional ones (OR 0.6, 95%CI 0.24–1.50). None of the isolates belonged to the other investigated CTX-M groups. The other most prevalent group, *bla*_SHV_, was found mainly in antibiotic-free (47%), followed by conventional type (16.7%) and organic type (8.5%). The proportion of SHV-1 isolates was different among the three groups (Pearson χ2: *p* < 0.001), with isolates from antibiotic-free farms having 4.48 OR (95%CI 1.84–10.92) of being SHV^+^ compared to conventional farms. The CIT and FOX enzymes were the most frequently observed plasmid-mediated AmpC-lactamases, but always in association with other enzymes belonging to ESBL-type. One out of twenty positive isolates for the CIT group was also positive for the CMY-2 group. 

Neither the proportion of isolates with an ESBL phenotype, AmpC phenotype, or being positive for TEM and CIT varied among the three categories (Pearson χ2: *p* > 0.05). 

### 3.4. Whole Genome Sequencing (WGS) 

A final assembly of the 13 sequences resulted in an average read quality after trimming of 34.79 (min 34.76; max 38.84) and a number of read pairs of 2,166,607 (min 1,859,398; max 3,042,982). The average number of contigs was 346 (min 174; max 714) with a mean length of 5,444,563 (min 5,126,897; max 5,872,105). The mean values for N50 and L50 were 214,872 (min 110,548; max 406,162) and 10 (min 5, max 18), respectively. 

Using the in silico method of ClermonTyping we identified three phylogroups: D, F and G. Seven isolates out of thirteen (53.8%) belonged to the phylogenetic group D, 4/13 (30.8%) to the G and two isolates were phylogenetic group F (15.4%). 

We identified nine STs and the most prevalent were ST117, counting for 30.8% (*n* = 4) of isolates, and ST349, counting for 15.4% (*n* = 2) of isolates. Other STs, with one isolate for each ST, are reported in Table 4. Nine isolates (69.3%) showed the APEC pathotype (Table 4). Among those, all isolates belonging to the ST117 and phylogenetic group G were APEC and ESBL, except one, which showed an APEC and AmpC phenotype (Figure 2). The assembled contigs were typed *in silico* using the CGE databases and SerotypeFinder predictions corroborated O and H antigens, shown in Table 4.

Among the 13 *E. coli* isolates, we described 30 genes coding for antibiotic resistance. The most frequent genes were: *mdf*(A) detected in all the tested isolates (100%), followed by *tet*(A) in 10/13 (77%), *aadA1* in 8 out of 13 (62%) while *bla*_CTX-M-1_ and *sul2* genes were present in 7/13 (54%) (Table 4). Moreover, we did not describe a relationship between the presence of AMR genes and the APEC pathotypes (Kruskal–Wallis rank sum test; *p* > 0.05). 

When we investigated the distribution of putative virulence genes in a subset of 13 isolates, we found 49 putative virulence genes and included them in the heat map (Figure 3). The most frequent virulence genes were *chuA*, *gad*, *iss*, *terC*, present in all isolates, while *traT* was detected in 12 out of 13 isolates (92.3%), *ompT* and *sitA* in 11/13 (84.6%), and *hlyF*, *iucC*, *iutA* in 10/13 (76.9%). Genes encoding for the heat-lable (LT) and heat-stable (ST) toxin, which characterize the Enterotoxigenic *E. coli* (ETEC) pathotype, were not detected in our isolates [47]. Putative virulence genes were not randomly distributed across the pathotypes (Kruskal–Wallis rank sum test; *p* = 0.01). The APEC isolates showed a higher number of virulence genes than the AFEC isolates (Figure 3). We did not observe differences in the presence of virulence factors between ESBL and AmpC phenotypes (Kruskal–Wallis rank sum test; *p* > 0.05).

## 4. Discussion

Resistance to extended-spectrum cephalosporins, molecules classified as HPCIAs by the WHO (WHO, 2019), in *E. coli* in the poultry sector has raised serious concerns regarding the transfer of clones to humans or the exchange of resistant genes between poultry and human flora. In the present study, the presence of ESBL/AmpC-EC in broilers at a slaughterhouse was observed in animals that were not subjected to antibiotic therapy during the production cycle. This confirms what has been described in the literature, where there was an expansion of ESBL/AmpC-EC clones along the broiler production cycle, even in the absence of selective pressure, despite low starting loads [48,49]. Resistance can be maintained, in the absence of antibiotic selection, through resistance mutations that may incur no fitness costs and compensate for the costs of resistance via second-site mutations, which restore organismal fitness [50].

The load of commensal *E. coli* was not even in the three types of production, with broilers from conventional farms having higher *E. coli* loads than O and AF broilers, although the difference among the three medians was small (approximately 0.2 Log CFU). This difference is likely due to the younger age at slaughter for C broilers compared to that of O and AF broilers [13], since a young age is associated with higher *E. coli* loads in the gut [51].

In our study, we found that the ESBL/AmpC-EC load in the caecum of broilers from conventional and antibiotic-free farms was approximately 10^4^/g, which was similar to that described in the literature for broilers of the same age [48,52]. 

In contrast, in O broilers, ESBL/AmpC-EC loads were more than ten times lower than those recorded in C broilers. Our data were obtained from samples collected from the caecum, and not from the meat. The final contamination of meat is generally lower than that of the gut. The contamination of meat depends on several factors apart from the presence of ESBL/AmpC-EC in the gut, including cross-contamination among carcasses, contamination by operators or from the premises of the food chain, and reduction of contamination after chilling [53]. According to the findings of the European official monitoring for AMR, the prevalence of ESBL/AmpC-EC in meat is lower, but still comparable to that in the gut, suggesting a strong connection between the contamination of the gut and that of meat in the broiler production chain [53]. Therefore, the low ESBL/AmpC-EC loads in the organic production system observed in this study may represent a reduced risk to the consumer [52].

ESBL/AmpC-EC isolates also showed resistance to multiple antibiotic classes historically used in the poultry sector, including penicillins, tetracyclines, quinolones, and sulfonamides, in agreement with what was observed in previous studies [16,54]. ESBL/AmpC-EC isolates from organic and antibiotic-free farms were less resistant to some antimicrobial classes. Among these, we observed minor resistance to antibiotics historically used in livestock, such as sulfonamides. Interestingly, the odds of being resistant to fluoroquinolones were halved in ESBL/AmpC-EC isolates from antibiotic-free farms compared to those from the conventional farms. The combined resistance to ESC and fluoroquinolones, two classes classified as HPCIA, is worrisome [13,55]. Our data suggest that isolates from antibiotic-free and organic broilers are less resistant than those from conventional farms. 

Biomolecular analyses have also revealed the presence of genes encoding TEM, SHV, and CTX-M in ESBL/AmpC-EC, confirming their involvement in the poultry sector [55]. In particular, CTX-M-1 is generally found on conjugative plasmids, is associated with other resistance genes, and is prevalent as the CTX-M-15 variant. CTX-M-15 has undergone global dissemination over the last 30 years [55,56]. This has been attributed to several factors, including the dispersion of mobile genetic elements and/or the dissemination of successful bacterial clones, favoured by the low fitness cost associated with the presence of CTX-M-1 genes [55]. In the veterinary sector, selective pressure generated by antibiotics is likely to have contributed to the spread of CTX-M-1 ESBL [55]. In our study, we observed a reduced proportion of CTX-M-1 in ESBL *E. coli* isolated from antibiotic-free broilers. This result might be explained by the presence of different bacterial clones on the antibiotic-free farms. Alternatively, the lack of selective pressure generated by antibiotic use may have limited the spread of CTX-M-1 among animals.

Taken together, our findings suggest that broilers from non-conventional farms, characterised by higher biosecurity standards, older age at slaughter, and no antibiotic use, are less risky in terms of ESBL/AmpC-EC contamination as compared to broilers from conventional farms.

The A phylogenetic group, associated with commensal bacteria in humans, is the most represented among ESBL/AmpC-EC of avian origin [57,58]. Numerous isolates from human extra-intestinal infections belong to the F and D phylogroups described in our study, together with commensal phylogroups [58]. Using WGS, we analysed the presence of *E. coli* isolates belonging to the G phylogroup, a new phylogenetic group generally misidentified as the F phylogenetic group through PCR analysis [59]. In humans, phylogenetic group G strains represent approximately 1% of the *E. coli* isolates and are found in both commensal and extra-intestinal pathogenic conditions, including septicaemia [60]. A limitation of our study is that only a small fraction of isolates underwent WGS. Nevertheless, we were able to describe the APEC pathotypes in our dataset independent of the production systems. APEC strains utilize different virulence and pathogenesis factors to cause disease in broilers, primarily adhesins, invasins, protectins, iron acquisition systems, and toxins [9], determinants that we found in our isolates. These factors facilitate the adhesion, invasion, evasion from host immune responses, colonisation, proliferation, and systemic dissemination of APEC, thereby allowing the establishment of infection in broilers [61]. Despite this classification, a considerable overlap of virulence determinants with human ExPEC can be found among a subset of APEC, indicating the non-host specificity of these strains and underscoring their zoonotic potential [9]. ExPEC subtypes, such as APEC isolates, can asymptomatically colonise the gut of a fraction of healthy animal population and survive in extra-intestinal environments, causing diseases in animals and humans throughout the food chain [4]. 

Consistently, phylogroup G strains belonged to ST117, which is the most prevalent phylogroup G lineage in broilers and poultry meat products from Northern Europe and Canada [62,63]. This sequence type has spread throughout the Nordic broiler production and has been implicated in large outbreaks of colibacillosis [63]. In our study, four ST117 strains were classified as APEC, confirming their pathogenic potential in poultry [64,65]. As part of ST117, we found another relevant ST associated with extra-intestinal infections, namely the ST362, with increased antibiotic resistance and enhanced virulence [66]. Thus, our data confirmed that ESBL/AmpC-EC in poultry could be APEC pathogens. These data confirm the possible role of the broilers as an ESBL/AmpC-EC reservoir associated with extra-intestinal forms in humans. In this study, broilers from non-conventional farms showed decreased ESBL/AmpC-EC loads at the slaughterhouse. As previously reported, ESBL/AmpC-EC isolates from non-conventional farms showed a more favourable antibiotic resistance profile than those from conventional farms. Overall, our study confirmed that antibiotic-free and organic production may contribute to a reduction in consumer exposure to ESBL/AmpC-EC.

## Figures and Tables

**Figure 1 antibiotics-11-01484-f001:**
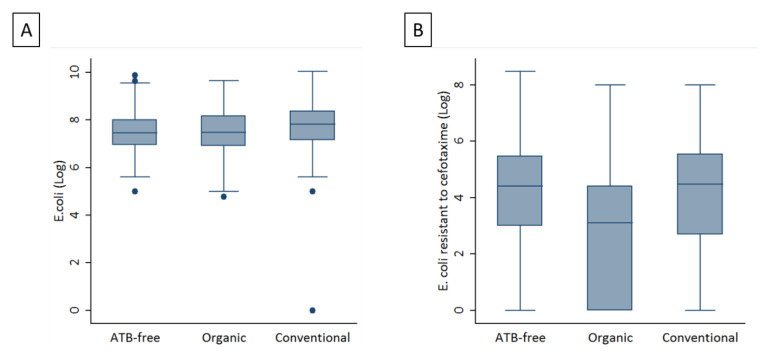
The averages of *E. coli* loads (Log CFU/g) (**A**) and averages of ESBL/AmpC-EC loads (Log CFU/g) (**B**) in the three production systems.

**Figure 2 antibiotics-11-01484-f002:**
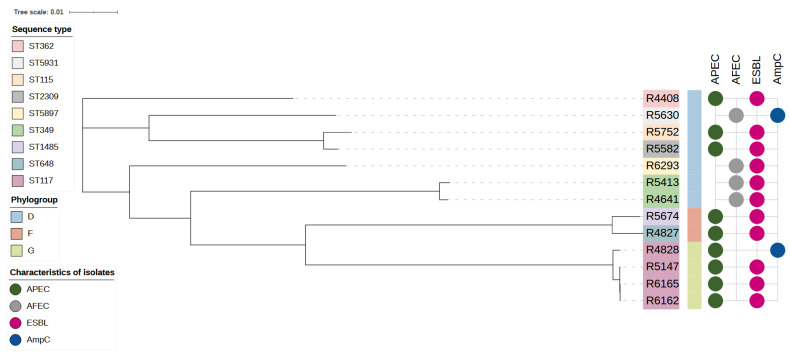
A phylogenetic tree containing 13 *Escherichia coli* isolates from poultry. The tree was inferred by using the iTOL interactive user interface (https://itol.embl.de accessed on 9 November 2021). Shading over tip labels indicates sequence types. The phylogenetic group of each isolate is also shown. The characteristic of isolates (APEC, AFEC, ESBL and AmpC) are annotated by colors and round shapes.

**Figure 3 antibiotics-11-01484-f003:**
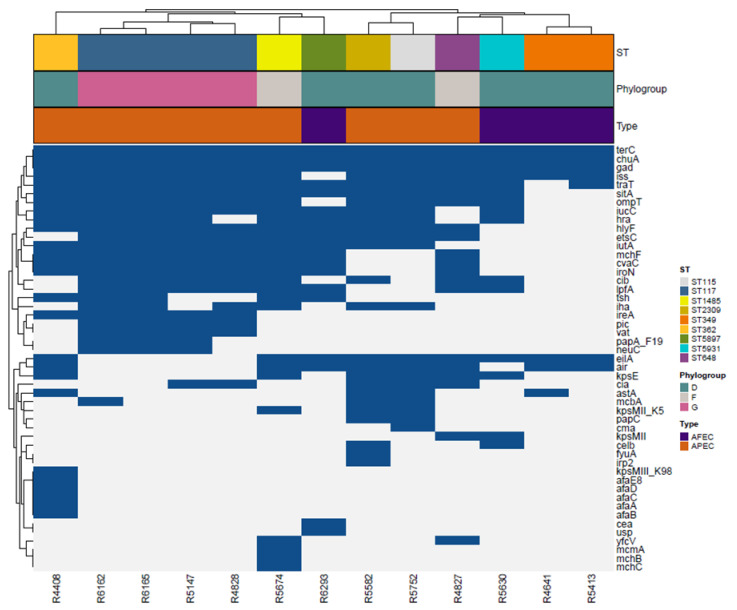
A heat map depicting virulence factors in the strain collection. The dendogram on the top represents clustering of *E. coli* isolates according to their sequence type (ST), phylogenetic group and the following characteristics: Avian Pathogenic *Escherichia coli* (APEC) in orange and Avian Faecal *Escherichia coli* (AFEC) in purple. STs and phylogroups are colour-coded as described in the legend.

**Table 1 antibiotics-11-01484-t001:** Antibiotic susceptibility in ESBL/AmpC-EC from antibiotic-free, organic and conventional farms. Percentages are shown in brackets.

Antibiotic Molecules *	Antibiotic-Free			Organic			Conventional		
	R	S	Tot	R	S	Tot	R	S	Tot
**AMP**	55 (100)	0 (0.00)	55 (100)	46 (97.87)	1 (2.13)	47 (100)	54 (100)	0 (0.00)	54 (100)
**AMC**	25 (45.45)	30 (54.54)	55 (100)	31 (65.96)	16 (34.04)	47 (100)	35 (64.81)	19 (35.19)	54 (100)
**CTX**	26 (47.27)	29 (52.73)	55 (100)	28 (59.57)	19 (40.43)	47 (100)	39 (72.22)	15 (27.78)	54 (100)
**KZ**	54 (98.18)	1 (1.82)	55 (100)	44 (93.62)	3 (6.38)	47 (100)	54 (100)	0 (0.00)	54 (100)
**C**	17 (30.91)	38 (69.09)	55 (100)	5 (10.64)	42 (89.36)	47 (100)	18 (33.33)	36 (66.67)	54 (100)
**CIP**	10 (18.18)	45 (81.82)	55 (100)	21 (44.68)	26 (55.32)	47 (100)	19 (35.19)	35 (64.81)	54 (100)
**CN**	3 (5.45)	52 (94.55)	55 (100)	4 (8.51)	43 (91.49)	47 (100)	3 (5.56)	51 (94.44)	54 (100)
**K**	4 (7.27)	51 (92.73)	55 (100)	1 (2.13)	46 (97.87)	47 (100)	9 (16.67)	45 (83.33)	54 (100)
**NA**	28 (50.91)	27 (49.09)	55 (100)	29 (61.70)	18 (38.30)	47 (100)	26 (48.15)	28 (51.85)	54 (100)
**S**	21 (38.18)	34 (61.82)	55 (100)	4 (8.51)	43 (91.49)	47 (100)	12 (22.22)	42 (77.78)	54 (100)
**S3**	38 (69.09)	17 (30.91)	55 (100)	27 (57.45)	20 (42.55)	47 (100)	47 (87.04)	7 (12.96)	54 (100)
**TE**	35 (63.64)	20 (36.36)	55 (100)	29 (61.70)	18 (38.30)	47 (100)	32 (59.26)	22 (40.74)	54 (100)
**SXT**	21 (38.18)	34 (61.82)	55 (100)	14 (29.79)	33 (70.21)	47 (100)	32 (59.26)	22 (40.74)	54 (100)

* Ampicillin (AMP), amoxicillin/clavulanic acid (AMC), cefotaxime (CTX), cefazolin (KZ), chloramphenicol (C), ciprofloxacin (CIP), gentamicin (CN), kanamycin (K), nalidixic acid (NA), streptomycin (S), sulfonamides (S3), tetracycline (TE) and sulfamethoxazole/trimethoprim (SXT).

**Table 2 antibiotics-11-01484-t002:** Results of a univariate analysis, taking conventional data as baseline values (OR = 1).

Antibiotic Molecules *	Production System	OR	95% CI	*p*-Value
**C**	Conventional	1	-	-
	Antibiotic-free	0.89	0.40, 2.00	0.786
	Organic	0.24	0.08, 0.71	0.007
**CIP**	Conventional	1	-	-
	Antibiotic-free	0.41	0.17, 0.99	0.045
	Organic	1.49	0.67, 3.32	0.33
**S3**	Conventional	1	-	-
	Antibiotic-free	0.33	0.13, 0.89	0.024
	Organic	0.20	0.08, 0.54	<0.001
**SXT**	Conventional	1	-	-
	Antibiotic-free	0.42	0.20, 0.92	0.028
	Organic	3.21	0.82, 12.50	0.082

* Chloramphenicol (C), ciprofloxacin (CIP), streptomycin (S), sulfonamides (S3) and sulfamethoxazole/trimethoprim (SXT).

**Table 3 antibiotics-11-01484-t003:** The distribution of resistant genes in ESBL/AmpC isolates. Percentages are shown in brackets.

Genotype	ESBL	AmpC	ESBL/AmpC	NEITHER
**CTXM-1**	56 (42.7)	0 (0)	0 (0)	0 (0)
**CTX-M-1/TEM**	38 (29)	1 (5.6)	0 (0)	2 (50)
**CTX-M-1/SHV**	2 (1.6)	0 (0)	0 (0)	0 (0)
**CTX-M-1/TEM/SHV**	2 (1.6)	0 (0)	0 (0)	0 (0)
**TEM**	0 (0)	0 (0)	0 (0)	0 (0)
**TEM/SHV**	7 (5.3)	0 (0)	0 (0)	0 (0)
**SHV**	26 (19.8)	0 (0)	0 (0)	1 (25)
**CIT/FOX/TEM**	0 (0)	11 (61.1)	0 (0)	0 (0)
**CIT/FOX**	0 (0)	3 (16.6)	1 (33.3)	0 (0)
**CIT/TEM**	0 (0)	2 (11.1)	1 (33.3)	0 (0)
**CIT/FOX/TEM/SHV**	0 (0)	1 (5.6)	0 (0)	0 (0)
**CTX-M-1/CIT**	0 (0)	0 (0)	1 (33.3)	0 (0)
**NONE**	0 (0)	0 (0)	0 (0)	1 (25)
**Total**	131 (100)	18 (100)	3 (100)	4 (100)

**Table 4 antibiotics-11-01484-t004:** Genomic characteristics of the 13 *Escherichia coli* isolates investigated using the whole genome sequencing and belonging from three productive systems.

Id	Productive System	Esbl	Ampc	Phlo-Group	St	Serotype	Type	Mutation	Plasmid	Resistance Genes	Virulence Factors
R4827	AF	POS	NEG	F	ST648	O83:H42	APEC		IncFIB(AP001918), IncFII, IncI1-I(Gamma), p0111	*tet*(A), *dfrA17*, *mdf*(A), *bla*_CTX-M-1_, *sul2*, *aadA5*	*air, chuA, cia, cib, cvaC, eilA, etsC, gad, hlyF, iroN, iss, kpsE, kpsMII, lpfA, mchF, ompT, sitA, terC, traT, yfcV*
R4828	AF	NEG	POS	G	ST117	O8:H4	APEC		IncB/O/K/Z, IncFIB(AP001918), IncFII, p0111	*aadA1*, *aac*(3)-VIa, *bla*_CMY-2_, *mdf*(A), *sul1*, *tet*(A)	*chuA, cia, cib, cvaC, etsC, gad, hlyF, iha, ireA, iroN, iss, iucC, iutA, lpfA, mchF, ompT, pic, sitA, terC, traT, vat*
R5147	AF	POS	NEG	G	ST117	H4	APEC	*gyrA* p.S83L	IncFIB(AP001918), IncFII, IncI1-I(Gamma)	*sul3*, *sul2*, *aadA2b*, *aadA1*, *mdf*(A), *tet*(A), *cmlA1*, *bla*_SHV-1_2	*chuA, cia, cib, cvaC, etsC, etsC, gad, hlyF, hra, ireA, iroN, iss, iucC, iutA, lpfA, mchF, neuC, ompT, papA_F19, pic, sitA, terC, traT, vat*
R6162	AF	POS	NEG	G	ST117	H4	APEC	gyrA p.S83L	IncFIB(AP001918), IncFIC(FII), IncFII, IncI1-I(Gamma)	*tet*(A), *cmlA1*, *bla*_SHV-12_, *mdf*(A), *sul3*, *sul2*, *aadA2b*, *aadA1*, *aph(3″)-Ib*, *aph(6)-Id*	*chuA, cib, cvaC, etsC, gad, hlyF, hra, iha, ireA, iroN, iss, iucC, iutA, lpfA, mcbA, mchF, neuC, ompT, papA_F19, pic, sitA, terC, traT, tsh, vat*
R6165	AF	POS	NEG	G	ST117	H4	APEC	gyrA p.S83L	IncFIB(AP001918), IncFIC(FII), IncI1-I(Gamma)	*mdf*(A), *tet*(A), *bla*_SHV-12_, *aph(3″)-Ib*, *aadA2b*, *aadA1*, *aph(6)-Id*, *sul3*, *sul2*, *cmlA1*	*chuA, cib, cvaC, etsC, gad, hlyF, hra, iha, ireA, iroN, iss, iucC, iutA, lpfA, mchF, neuC, ompT, papA_F19, pic, sitA, terC, traT, tsh, vat*
R4408	C	POS	NEG	D	ST362	O15:H1	APEC	gyrA p.S83L	IncFIB(AP001918), IncFII, IncI1-I(Gamma)	*fosX*, *mph*(B), *mdf*(A), *aadA1*, *aph(6)-Id*, *aph(3″)-Ib*, *bla*_CTX-M-1_, *tet*(A), *sul1*, *catA1*, *dfrA1*	*afaA, afaB, afaC, afaD, afaE8, air, astA, chuA, cvaC, eilA, gad, hlyF, hra, ireA, iroN, iss, iucC, iutA, kpsE, kpsMIII_K98, mchF, ompT, sitA, terC, traT, tsh*
R4641	C	POS	NEG	D	ST349	O166:H15	AFEC	gyrA p.S83L	IncI1-I(Gamma), p0111	*bla*_CTX-M-1_, *mdf*(A)	*air, astA, chuA, eilA, gad, iss, terC*
R5413	C	POS	NEG	D	ST349	H15	AFEC	gyrA p.S83L	IncFII(29), IncI1-I(Gamma), p0111	*sul2*, *aph(3″)-Ib*, *aph(6)-Id*, *dfrA14*, *bla*_CTX-M-1_, *mdf*(A)	*air, chuA, eilA, gad, iss, terC, traT*
R5582	C	POS	NEG	D	ST2309	O15:H6	APEC		Col156, IncFIB(AP001918), IncFII, IncI1-I(Gamma), IncN, IncY	*aadA2b*, *aadA1*, *sul3*, *mdf*(A), *tet*(A), *cmlA1*, *bla*_TEM-1B_, *bla*_SHV-12_, *qnrS1*	*air, astA, celb, chuA, ciacib, eilA, etsC, fyuA, gad, hlyF, hra, iha, irp2, iss, iucC, iutA, kpsE, kpsMII_K5, mcbA, ompT, papC, sitA, terC, traT*
R5752	C	POS	NEG	D	ST115	O50:H6	APEC		Col(pHAD28), IncB/O/K/Z, IncFIB(AP001918), IncFII, IncI1-I(Gamma), p0111	*aadA5*, *bla*_CTX-M-1_, *mdf*(A), *sul2*, *qnrB19*, *dfrA17*	*air, astA, chuA, cia, cma, eilA, etsC, gad, hlyF, hra, iha, iss, iucC, iutA, kpsE, kpsMII_K5, mcbA, ompT, papC, sitA, terC, traT*
R6293	C	POS	NEG	D	ST5897	H31	AFEC		IncFIA(HI1), IncFIB(AP001918), IncFIC(FII), IncFII(pHN7A8), IncI1-I(Gamma), IncX1	*tet*(A), *catA1*, *cmlA1*, *mdf*(A), *aadA1*, *aph(3′)-Ia*, *aadA2b*, *sul3*, *bla*_TEM-106_, *bla*_TEM-126_, *bla*_CTX-M-1_, *bla*_TEM-220_, *bla*_TEM-1B_, *bla*_TEM-135_,	*air, cea, chuA, cvaC, eilA, etsC, gad, hlyF, hra, iroN, iucC, iutA, lpfA, mchF, sitA, terC, traT, tsh, usp*
R5630	O	NEG	POS	D	ST5931	O1:H1	AFEC	gyrA p.S83L	Col156, Col8282, IncB/O/K/Z, IncI2(Delta)	*sul1*, *bla*_CMY-2_, *bla*_TEM-1C_, *mdf*(A), *aac(3)-VIa*, *aadA1*, *dfrA1*, *tet*(A)	*celb, chuA, cib, eilA, gad, hra, iha, iss, iucC, kpsE, kpsMII, lpfA, ompT, sitA, terC, traT*
R5674	O	POS	NEG	F	ST1485	O83:H42	APEC	gyrA p.S83L, gyrA p.D87Y, parC p.S80I	IncFIA, IncFIB(AP001918), IncFIC(FII), IncI1-I(Gamma)	*bla*_CTX-M-1_, *bla*_TEM-1B_, *tet*(A), *mdf*(A), *sul2*, *aph(6)-Id*, *aph(3″)-Ib*, *dfrA14*	*air, chuA, cib, cvaC, eilA, etsC, gad, hlyF, hra, iha, iroN, iss, iucC, iutA, kpsE, kpsMII_K5, lpfA, mchB, mchC, mchF, mcmA, ompT, sitA, terC, traT, tsh, yfcV*

## Data Availability

The raw sequencing data has been submitted to NCBI’s Sequence Read Archive (SRA) repository (BioProject: PRJNA882337; Biosample: SUB9357225, accessions SAMN30930178 to SAMN30930190).

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
