# Peer review of "Assessing the Load, Virulence and Antibiotic-Resistant Traits of ESBL/Ampc E. coli from Broilers Raised on Conventional, Antibiotic-Free, and Organic Farms"

_antibiotics, 2022, doi:10.3390/antibiotics11111484_

Round 1

Reviewer 1 Report

General remarks

1.     The authors present the comparison of load, virulence and antibiotic-resistant characteristics (antibiotic susceptibility, detection of resistance genes, whole genome sequencing) of ESBL/AmpC E. coli strains isolated from chicken raised on conventional, antibiotic-free and organic farms.

2.     The topic of antibiotic resistance especially ESBL producing bacteria is a hot issue because of the threat of spread of antibacterial resistance, it is a very important One Health question. The paper fits into the journal of Antibiotics.

3.     The paper is a well written one, the hypothesis of the authors is clear, they used appropriate methods to answer the questions, the results and the conclusions are correct.

4.     The style of the paper is good, it is easy to read, its English is good.

Specific remarks

1.      ESBL/pAmpC is mentioned in the abstract without explanation (Line 18). It is explained only in the Materials and methods chapter (Lines123-125).

2.      The name of E.coli must be written in italics both in the text and in the list of references (Lines 18, 28, 29; 438-602).

3.      The word “have” is doubled (Line 19).

4.      Cell counts should be given in numbers, too, not only in log values (Line 23).

5.      ..antibiotics than … Separate the two words! (Line 26).

6.      The name of E. coli is singular and not plural (Lines 37, 50).

7.      Cited literature has to be given into the list of references, not in the text (Lines 67-68; 165-166).

8.      Production systems AF, O and C have to be explained also in the Materials and methods chapter, not only in the abstract (Lines 86-87).

9.      Producers of equipment have to be given (Line 92).

10.   10-1 and 10-8 are upper indices (Line 94).

11.   The authors should give how the total number of E. coli was measured (Lines 90-108).

12.   The word “phylogenetic” need not be capitalised (Line 138).

13.   The difference between the groups shown on Fig. 1. A does not seem to be significant. Would you please comment it (Page 5)?

14.   “Distribution of resistance genes” is recommended (Line 272).

15.   Complete bibliographic data are needed for references 17, 18, 20, 23.

16.   Titles of papers on the list of references need not be capitalised (Lines 438-602).

Conclusion

The paper is recommended for publication with minor modifications.

Author Response

Reviewer 1

General remarks

The authors present the comparison of load, virulence and antibiotic-resistant characteristics (antibiotic susceptibility, detection of resistance genes, whole genome sequencing) of ESBL/AmpC E. coli strains isolated from chicken raised on conventional, antibiotic-free and organic farms.

The topic of antibiotic resistance especially ESBL producing bacteria is a hot issue because of the threat of spread of antibacterial resistance, it is a very important One Health question. The paper fits into the journal of Antibiotics.

The paper is a well written one, the hypothesis of the authors is clear, they used appropriate methods to answer the questions, the results and the conclusions are correct.

The style of the paper is good, it is easy to read, its English is good.

Authors: We thank the reviewer for the constructive comments. We have modified the paper according to the suggestions.

Specific remarks

  1. ESBL/pAmpC is mentioned in the abstract without explanation (Line 18). It is explained only in the Materials and methods chapter (Lines123-125).

Authors: we have included the complete name on the line 18, as suggested.

  1. The name of coli must be written in italics both in the text and in the list of references (Lines 18, 28, 29; 438-602).

Authors: done.

  1. The word “have” is doubled (Line 19).

Authors: we thank the reviewer and we apologize for the mistake. We have modified it accordingly.

  1. Cell counts should be given in numbers, too, not only in log values (Line 23).

Authors: we included the data either in the abstract or in results section.

  1. ..antibiotics than … Separate the two words! (Line 26).

Authors: done.

  1. The name of coli is singular and not plural (Lines 37, 50).

Authors: we have included the word “isolates” after the E. coli

  1. Cited literature has to be given into the list of references, not in the text (Lines 67-68; 165-166).

Authors: done, thanks.

  1. Production systems AF, O and C have to be explained also in the Materials and methods chapter, not only in the abstract (Lines 86-87).

Authors: done.

  1. Producers of equipment have to be given (Line 92).

Authors: added.

  1. 10-1 and 10-8 are upper indices (Line 94).

Authors: done.

  1. The authors should give how the total number of coli was measured (Lines 90-108).

Authors: we added this information in the text.

  1. The word “phylogenetic” need not be capitalised (Line 138).

Authors: done, thanks.

  1. The difference between the groups shown on Fig. 1. A does not seem to be significant. Would you please comment it (Page 5)?

Authors: using the statistical methods reported in the materials and methods, we described a difference among the three groups. We hypothesized that the difference could be assigned to the conventional vs. ATB-free productive systems.

  1. “Distribution of resistance genes” is recommended (Line 272).

Authors: modified, thanks.

  1. Complete bibliographic data are needed for references 17, 18, 20, 23.

Authors: done.

  1. Titles of papers on the list of references need not be capitalised (Lines 438-602).

Authors: we modify the reference section. We thank the reviewer for the suggestion.

Reviewer 2 Report

Dear authors, your research is impressively comprehensive, in my opinion well-chosen, presented methods, beautifully processed and interpreted results and very current and important topics. To the best of my ability, I do not have a single recommendation to improve the manuscript.

Author Response

Authors: we thank the reviewer for the comment.

Reviewer 3 Report

Comments to the manuscript; antibiotics-1956432

 Assessing the load, virulence and antibiotic-resistant traits of ESBL/AmpC E. coli from broilers raised on conventional, antibiotic-free, and organic farms

Specific comments to the manuscript:

Abstract:

Line 18: E. coli must be italicized

Line 19: Remove have (repeated)

Line 21: Replace was …..with were (Isolates were)

Line 24: Only mention the p value upto three decimal places (e.g., p<0.001)

Line 28: Same as line 18 (E. coli must be italicized)

Introduction:

Line 56-57: Some of the important virulence factors may be enumerated or briefly added in introduction part.

Line 81-82: Similar to above comment ST (Sequence type) previously observed in the region or most commonly encountered sequence type can be added in introduction section in one or two sentence.

Materials and methods:

Line 89: Bacteriological culture can be used as a subheading instead of microbiological culture.  Line 111: Be careful with spelling of Mueller Hinton Agar. Please correct the same.

Line 113: Add quality control.

Line 122: Besides using the reference for criterion, we may briefly describe the criterion here for easy understanding of the reader and to maintain flow of reading.

Line 123-125: The abbreviated forms used here were for E. coli, but again used for Enterobacteriaceae here. Please correct or restructure. The authors may consider briefly describing DDST method of phenotypic detection as there are various modifications to this method in various literatures. The antibiotics used for the DDST method may be described.

Line 179-180: APEC genotypes were described when at least three among iroN, iutA, iss, ompT, and hlyF genes were present [Johnson et al. 2008]. The criterion described here voilates the original criterion described by Johnson et al. 2008, which necessiates the presence of four or more above stated virulence genes for defining the pathotype/genotype to be APEC. Please rectify or restruture the same or may be rectified or restructured in the results also. The similar methods/criterion were also used by other authors for APEC characterization viz., Aawdeh, 2018; Kazibwe et al. 2020; Grakh et al. 2022. These can be considered for rectifying the methods section.

Results:

Line 233: classified as MDR, because .....of resistance

Line 286-287: The APEC should be defined using the original criterion by Johnson et al. (2008), by the presence of four or more virulence genes out of a set of five (hlyF, ompT, iss, iroN and iutA). If the authors wish to propose new criterion they may not use the reference for Johnson et al. (2008). The presence of these viulence genes may vary among countries and continent as has been described in various studies around the world for instance, India, Brazil, Australia etc.

Line 3019-310 and Table 4. The isolate ID: R4828 has all the five virulence genes for the definition of APEC as per criterion used and writting defined as APEC, but an APEC isolate may not be non-ExPEC as APEC itself is a subpathotype of ExPEC. The criterion used for ExPEC characterisation need to be revisited or corrected ?

Line 318 and Fig.3 : The heat map depicts that some of the isolates defined as ExPEC in the current study were later included in the APEC (which is well understood) and some under AFEC. The distribution of ExPEC to AFEC may not be justifiable as AFEC are not subtype of ExPEC like APEC. AFEC are commensal or microbiome of poultry and are not responsibel for any pathogenesis or disease (generally). So, this section needs some revision according to me. The criterion used for defining ExPEC may be revised or revisited again. These isolates should not be into AFEC. Similarly, the non-ExPEC should not be APEC ?

Discussion:

Line 415-418: The discussion on ExPEC and APEC may be elobaorated further for beteer understanding of the readers.  Please elaborate or discuss about the elt gene in brief. What is the importance of the gene. Why this gene was targetted and which other genes are generlly or commonly employed for ETEC detection and confirmation by molecualr mehods. 

Was there any association tested for the presence of virulence genes and ESBL/AmpC? If yes what was the observed association/diffrence? It may be added as ceratin previous studies have mentioned the same and revealed several interesting findings. 

General comment:

The study is much informative with appropriate number of samples and apt sampling method. The study revealed the important findings that organic broilers revealed significantly less load of ESBL EC which is a relief and an interesting finding to promote this method of broiler farming to other parts of world. The work is extensive with regards to WGS, molecular and phenotypic detection of ESBL, characterization of ExPEC and APEC. However, the section on criterion used for defining ExPEC and APEC needs restructuring or careful observations as I have mentioned in the comments. The manuscript may be improved with the addition/resolution of above said comments.   

Author Response

Reviewer 3

General comment:

The study is much informative with appropriate number of samples and apt sampling method. The study revealed the important findings that organic broilers revealed significantly less load of ESBL EC which is a relief and an interesting finding to promote this method of broiler farming to other parts of world. The work is extensive with regards to WGS, molecular and phenotypic detection of ESBL, characterization of ExPEC and APEC. However, the section on criterion used for defining ExPEC and APEC needs restructuring or careful observations as I have mentioned in the comments. The manuscript may be improved with the addition/resolution of above said comments.  

Authors: We thank the reviewer for the constructive comments. We have modified the paper according to your suggestions.

Specific comments to the manuscript:

Abstract:

Line 18: E. coli must be italicized

Authors: done, thanks.

Line 19: Remove have (repeated)

Authors: done, thanks.

Line 21: Replace was …..with were (Isolates were)

Authors: modified.

Line 24: Only mention the p value upto three decimal places (e.g., p<0.001)

Authors: we modified it in all the text, thanks.

Line 28: Same as line 18 (E. coli must be italicized)

Authors: done, thanks.

Introduction:

Line 56-57: Some of the important virulence factors may be enumerated or briefly added in introduction part.

Authors: we have added sentences about virulence factors in APEC pathotype.

Line 81-82: Similar to above comment ST (Sequence type) previously observed in the region or most commonly encountered sequence type can be added in introduction section in one or two sentence.

Authors: we have added a sentence about ST in the introduction, as requested.

Materials and methods:

Line 89: Bacteriological culture can be used as a subheading instead of microbiological culture.

Authors: we have modified the subheading accordingly to your suggestion.

Line 111: Be careful with spelling of Mueller Hinton Agar. Please correct the same.

Authors: modified.

Line 113: Add quality control.

Authors: added.

Line 122: Besides using the reference for criterion, we may briefly describe the criterion here for easy understanding of the reader and to maintain flow of reading.

Authors: we added the interpretation criteria as suggested.

Line 123-125: The abbreviated forms used here were for E. coli, but again used for Enterobacteriaceae here. Please correct or restructure. The authors may consider briefly describing DDST method of phenotypic detection as there are various modifications to this method in various literatures. The antibiotics used for the DDST method may be described.

Authors: we added a brief description as suggested.

Line 179-180: APEC genotypes were described when at least three among iroN, iutA, iss, ompT, and hlyF genes were present [Johnson et al. 2008]. The criterion described here voilates the original criterion described by Johnson et al. 2008, which necessiates the presence of four or more above stated virulence genes for defining the pathotype/genotype to be APEC. Please rectify or restruture the same or may be rectified or restructured in the results also. The similar methods/criterion were also used by other authors for APEC characterization viz., Aawdeh, 2018; Kazibwe et al. 2020; Grakh et al. 2022. These can be considered for rectifying the methods section.

Authors: we thank the reviewer for the comment. We have modified the criteria according to the recent publication of Johnson et al 2022 and consequently the text.

Results:

Line 233: classified as MDR, because .....of resistance

Authors: modified.

Line 286-287: The APEC should be defined using the original criterion by Johnson et al. (2008), by the presence of four or more virulence genes out of a set of five (hlyF, ompT, iss, iroN and iutA). If the authors wish to propose new criterion they may not use the reference for Johnson et al. (2008). The presence of these viulence genes may vary among countries and continent as has been described in various studies around the world for instance, India, Brazil, Australia etc.

Authors: we thank the reviewer for the comment. We have modified the criteria according to the recent publication of Johnson et al 2022 and consequently updated the information in the text.

Line 3019-310 and Table 4. The isolate ID: R4828 has all the five virulence genes for the definition of APEC as per criterion used and writting defined as APEC, but an APEC isolate may not be non-ExPEC as APEC itself is a subpathotype of ExPEC. The criterion used for ExPEC characterisation need to be revisited or corrected ?

Authors: we thank the reviewer for the comment. We have modify the criteria for APEC pathotype and modified the content of the table 4.

Line 318 and Fig.3 : The heat map depicts that some of the isolates defined as ExPEC in the current study were later included in the APEC (which is well understood) and some under AFEC. The distribution of ExPEC to AFEC may not be justifiable as AFEC are not subtype of ExPEC like APEC. AFEC are commensal or microbiome of poultry and are not responsibel for any pathogenesis or disease (generally). So, this section needs some revision according to me. The criterion used for defining ExPEC may be revised or revisited again. These isolates should not be into AFEC. Similarly, the non-ExPEC should not be APEC ?

Authors: we have modify the heat map and the text accordingly to the updated classification of our isolates thank to your previous comment.

Discussion:

Line 415-418: The discussion on ExPEC and APEC may be elobaorated further for beteer understanding of the readers.  Please elaborate or discuss about the elt gene in brief. What is the importance of the gene. Why this gene was targetted and which other genes are generlly or commonly employed for ETEC detection and confirmation by molecualr mehods. 

Authors: we thank the reviewer for the opportunity to discuss this point. As stated above, we completely revised our classification of E.coli pathotypes according to the recent study by Johnson et al. We did not find genes characterizing the ETEC pathotype in our collection. As requested by the reviewer, we added a sentence clarifying this point.

Was there any association tested for the presence of virulence genes and ESBL/AmpC? If yes what was the observed association/diffrence? It may be added as ceratin previous studies have mentioned the same and revealed several interesting findings. 

Authors: we have tested it but we did not described any difference (Kruskal-Wallis rank sum test; p-value= 0.27). As suggested we have included this result in our paper.
